# The North American bullfrog draft genome provides insight into hormonal regulation of long noncoding RNA

S. Austin Hammond[1], René L. Warren[1], Benjamin P. Vandervalk[1], Erdi Kucuk[1], Hamza Khan[1], Ewan A. Gibb[1], Pawan Pandoh[1], Heather Kirk[1], Yongjun Zhao[1], Martin Jones [1], Andrew J. Mungall [1], Robin Coope[1], Stephen Pleasance[1], Richard A. Moore[1], Robert A. Holt[1], Jessica M. Round[2], Sara Ohora[2], Branden V. Walle[2], Nik Veldhoen[2], Caren C. Helbing[2] & Inanc Birol [1]

Frogs play important ecological roles, and several species are important model organisms for scientific research. The globally distributed Ranidae (true frogs) are the largest frog family, and have substantial evolutionary distance from the model laboratory *Xenopus* frog species. Unfortunately, there are currently no genomic resources for the former, important group of amphibians. More widely applicable amphibian genomic data is urgently needed as more than two-thirds of known species are currently threatened or are undergoing population declines. We report a 5.8 Gbp (NG50 = 69 kbp) genome assembly of a representative North American bullfrog (*Rana [Lithobates] catesbeiana*). The genome contains over 22,000 predicted protein-coding genes and 6,223 candidate long noncoding RNAs (lncRNAs). RNA-Seq experiments show thyroid hormone causes widespread transcriptional change among protein-coding and putative lncRNA genes. This initial bullfrog draft genome will serve as a key resource with broad utility including amphibian research, developmental biology, and environmental research.

[1] Canada's Michael Smith Genome Sciences Centre, BC Cancer Agency, 570 West 7th Ave - Suite 100, Vancouver, BC, Canada V5Z 4S6. [2] Department of Biochemistry and Microbiology, University of Victoria, Petch Bldg Room 207, 3800 Finnerty Road, Victoria, BC, Canada V8P 5C2. Correspondence and requests for materials should be addressed to C.C.H. (email: chelbing@uvic.ca) or to I.B. (email: ibirol@bcgsc.ca)

Living in the most varied environments with both aquatic and terrestrial life stages, frogs are known to be evolutionary innovators in responding to challenges. However, diseases and infections such as chytrid fungus[1], iridovirus[2], and trematode parasites[3] are causing local and regional die-offs. In tandem with habitat loss, which is exacerbated by climate change, these factors have resulted in a worldwide amphibian extinction event unprecedented in recorded history: over two-thirds of ~7,000 extant species are currently threatened or declining in numbers (http://amphibiaweb.org/declines/declines.html).

Frogs are important vertebrates, and have provided key discoveries in the fields of ecology, evolution, biochemistry, physiology, endocrinology, and toxicology[4]. Yet, there are considerable gaps in the data required to understand their basic biology at the molecular level; few frog genomes are available, and none represent a member of the Ranidae (true frogs), the largest frog family with species found on every continent except Antarctica. The North American bullfrog, *Rana* (*Lithobates*) *catesbeiana*, is an ideal species for building a representative Ranid genomic resource because it is consistently diploid, and has the widest global distribution of any true frog. Originally from eastern North America, the bullfrog has been introduced throughout the rest of North America, South America, Europe and Asia. It is farmed for food in many locations worldwide, and is considered an invasive species in several regions[5].

The genomes of two *Xenopus* species (*X. tropicalis* and *X. laevis*) have been sequenced and annotated[6,7], but these Pipids have an estimated divergence from the Ranidae ~260 million years ago (MYA)[8]. This evolutionary separation is accentuated by their differing life histories, behavior, and markedly different sex differentiation systems[9]; recent evidence suggests that the innate immune system of *Xenopus* is fundamentally different from three frog families including the Ranidae[10]. As a consequence, the degree of sequence variation is such that Ranid-specific genomic and transcriptomic data are required to satisfy the currently unmet need for these resources in Ranid studies[4]. The genome of a more-closely related frog, the Tibetan Plateau frog (*Nanorana parkeri*), has been recently released[11], though this species is also substantially separated from Ranids by ~89 MYA[12].

Using some of the latest sequencing and bioinformatics technologies, we have sequenced, assembled, and annotated an initial draft sequence of the ~ 5.8 billion nucleotide North American bullfrog genome (scaffold NG50 length 68,964 bp). We predict 52,751 transcripts from 42,387 genes, of which 22,204 have supporting biological evidence, and are deemed high confidence. We anticipate that this much-needed resource, which we make public alongside comprehensive transcriptome assembly data, will directly and immediately impact genetic, epigenetic, and transcriptomic true frog studies. On a wider scale, it will empower developmental biology research ranging from amphibians to mammals, provide opportunities for direly needed insights to curb rapidly declining Ranid populations, and further our understanding of frog evolution.

## Results

**Assembly and annotation.** The draft assembly of the *R. catesbeiana* genome consists of 5.8 Gbp of resolved sequence (Table 1). The majority of raw reads from the paired-end tag (PET) libraries were successfully merged (63–75%), yielding longer pseudo-reads (mean ± SD, 446±107 bp). The success of the pre-assembly read merging allowed us to use read-to-read overlap lengths (represented by the assembly parameter k) greater than our shortest PET read length, increasing our ability to resolve short repetitive sequences. Genome scaffolding with orthogonal data, which included our build of the reference

**Table 1 Assembly statistics for sequences 500 bp or more in length in the final assembly**

|  | Unitig | Contig | Scaffold |
|---|---|---|---|
| Number ≥ 500 bp | 2,737,303 | 2,191,947 | 1,533,531 |
| Number ≥ N50 | 420,964 | 295,271 | 24,788 |
| Number ≥ NG50 | 438,623 | 300,168 | 18,459 |
| N80 (bp) | 1,252 | 1,862 | 2,959 |
| N50 (bp) | 3,620 | 5,302 | 51,621 |
| NG50 (bp) | 3,509 | 5,239 | 68,964 |
| N20 (bp) | 8,198 | 11,740 | 194,549 |
| Max (bp) | 68,999 | 90,443 | 1,775,282 |
| Reconstruction (Gbp) | 5.715 | 5.787 | 5.843 |

See Methods section for details

transcriptome and scaffolds assembled at a lower k value, greatly improved the contiguity of the resulting assembly (Table 1). We assessed the improvement to the assembly after each round of scaffolding using the NG50 length metric and the number of complete and fragmented near-universal single-copy orthologs using BUSCO with its tetrapoda set, which reports a proxy metric for assembly completeness in the genic space[13]. Using the Synthetic Long-Reads (SLR) and the Kollector[14] targeted gene assembly (TGA) tool, RAILS[15] merged over 56 thousand scaffolds; this permitted the recovery of an additional 113 BUSCOs, and raised the contiguity of the assembly to ~30 kbp (Supplementary Table 1). The most dramatic improvements to assembly contiguity and resolved BUSCOs were obtained using LINKS[16] and the mate pair (MPET) reads (NG50 increase of ~16 kbp and 146 additional complete BUSCOs; Supplementary Table 1), followed by the combined Kollector TGA and the lower-k whole genome assembly (~8 kbp improvement to NG50 and 103 additional complete BUSCOs; Supplementary Table 1). Finally, ARCS[17] scaffolding using 10x Genomics Chromium linked reads yielded a nearly 11 kbp improvement in contiguity, measured by the NG50 length, and further recovery of 38 complete BUSCO genes (Supplementary Table 1).

The automated gap closing tool Sealer[18] was used three times during the assembly process. First, prior to the rounds of rescaffolding to increase the amount of resolved sequence available to inform the scaffolding algorithms, and then post-rescaffolding to improve the sequence contiguity and content for the MAKER2 gene prediction pipeline[19]. Sealer initially closed 55,657 gaps, and resolved nearly 9 Mbp of sequence in the initial scaffolds, and further resolved 20 Mbp of sequence in its second round, closing 61,422 additional gaps. A final 7,144 gaps were closed by Sealer and 1.9 Mbp of sequence resolved in the ARCS scaffolded draft assembly.

We identified > 60% of the *R. catesbeiana* genome as putative interspersed repeats (Supplementary Table 2). CEGMA, the better-known precursor to BUSCO, also gives a proxy for genome completeness based on 248 highly conserved "core eukaryotic genes"[20]. The draft bullfrog genome includes 101 (40.7%) "complete" CEGMA genes, 212 complete or fragmented CEGMA genes (85.5%), 1,789 (45.3%) "complete" BUSCO genes, and 2,647 (67.0%) complete or fragmented BUSCO genes. Application of the MAKER2 genome annotation pipeline to an earlier stage draft assembly (version 2, see Supplementary Methods for a description of assembly versions) resulted in a set of 42,387 predicted genes and 52,751 transcripts. The criteria applied to identify the high confidence set of genes reduced the population by approximately half, to 22,204 genes and 25,796 transcripts (Supplementary Fig. 1). Of this high confidence set, 15,122 predicted proteins encoded by 12,671 genes could be assigned a

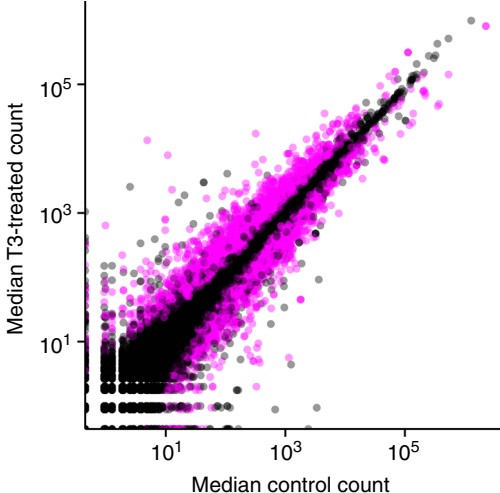

**Fig. 1** Median DESeq2-normalized counts of genes detected in the back skin of premetamorphic *Rana catesbeiana* tadpoles treated with vehicle control or T3 for 48 h. Gene transcripts determined to be significantly differentially expressed (DESeq2 adjusted *p*-value < 0.05) are indicated in pink, while the remainder are semi-transparent black to convey density. Both predicted protein coding and putative lncRNA genes are depicted

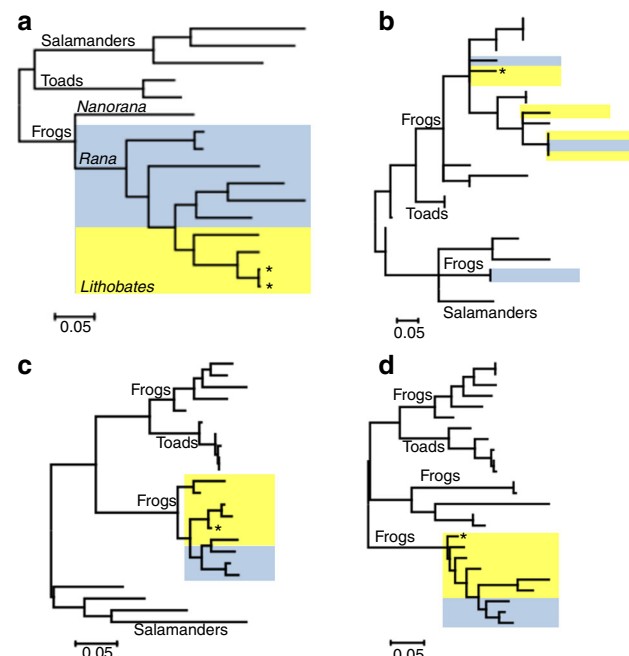

**Fig. 2** Molecular phylogenetic analysis of amphibian mitochondrial genomes and genes. The phylogenetic tree is drawn to scale, with branch lengths measured in the number of substitutions per site. The analysis involved **a** complete mitochondrial (Mt) genome sequences of salamanders, toads and frogs, classified as *Rana* (blue highlight) or *Lithobates* (yellow highlight). Analysis of Mt genes **b** *cyb*, **c** *rnr1*, and **d** *rnr2* of selected frog species. Position of *R. catesbeiana* indicated by an asterisk

functional annotation based on significant similarity to a SwissProt entry. An additional 3,668 proteins from 3,302 genes had significant similarity to a Pfam[21] domain and could accordingly be putatively annotated. Furthermore, 680 proteins from 590 genes were identified as particularly robust predictions by GeneValidator[22] (score ≥ 90). This "golden" set includes several members of the Homeobox (HOX), Forkhead box (FOX), and Sry-related HMG box (SOX) gene families, which are transcription factors involved in developmental regulation[23]. Immune-related genes, including interleukins 8 and 10, interferon gamma, and Toll-like receptors 3 and 4 were also confidently annotated.

**lncRNA prediction**. The discovery and analysis of lncRNAs represent a new frontier in molecular genetics, and is of major relevance to the biology behind this functional and largely unexplored component of the transcriptome. The low degree of lncRNA primary sequence conservation between organisms, and the lack of selective pressure to maintain ORF integrity or codon usage complicates traditional similarity-based discovery methods[24]. We employed a multilayer subtractive approach to lncRNA detection that relied on identifying and removing putative protein coding transcripts from the BART reference transcriptome and additional BART Jr. sequences (see Methods section for details). This selection strategy also demanded that candidate lncRNA sequences contain a polyadenylation signal, and be confidently aligned to the draft genome assembly (version 2, see Methods and Supplementary Methods for details). The final set of candidate lncRNA transcripts consisted of 6,223 sequences, which ranged in length from 200 bp to almost 11 kb, with a median length of 973 bp.

**Differential expression**. Characterization of the regulatory factors that mediate thyroid hormone (TH) dependent initiation of tissue specific gene expression programs during tadpole metamorphosis have been extensively studied in *X. laevis*. However, this species experiences markedly different environmental conditions in its natural habitat than many Ranids do, and these experiments employed supraphysiological levels of TH[25,26]. The assembled and annotated draft bullfrog genome offers the first

opportunity to study gene expression changes using a reference sequence that is directly relevant to Ranid species. Our present analysis of the TH-induced metamorphic gene expression program in the back skin detected nearly 5,000 protein coding genes significantly (*p* < 0.05) differentially expressed upon T3 (the TH 3,3′,5-triiodo-L-thyronine) treatment (Fig. 1), including those found previously through targeted quantitative polymerase chain-reaction (qPCR) experiments (Supplementary Table 3). The most prominent "biological process" gene ontologies associated with the Swiss-Prot derived functional annotations are related to RNA/DNA processing, signal transduction (including hormone signaling), and functions related to cell growth and division (Supplementary Fig. 2). A selection of new transcripts related to RNA/DNA processing were evaluated using qPCR, and found to show similar relative abundance as observed with the RNA-Seq data (Supplementary Fig. 3; Supplementary Table 3).

The effect of T3 treatment was not limited to the predicted protein coding genes, as expression of almost 1/6th of the candidate lncRNAs was also significantly affected. A selection of the 1,085 differentially expressed lncRNA transcripts was evaluated using qPCR (Supplementary Fig. 4).

**Amphibian phylogenetic analysis**. Frog taxonomy is subject of debate, particularly the proposition to transfer New World members of the genus *Rana* into the new genus *Lithobates*[27]. To address the controversy, we performed a number of phylogenetic experiments comparing selected amphibian mitochondrion (Mt) genomes and Mt genes at the nucleotide level (Fig. 2). As expected, we observe clear separation of salamanders and toads (genus *Bufo*) from other species as outgroups (Fig. 2a; Supplementary Figs. 5–8). We color-coded the *Lithobates* and *Rana* in yellow and blue, respectively, as re-classified by Frost et al.[28], to identify the relative genus positioning within the generated

**Table 2 ABySS-Bloom sequence identity calculations between amphibian genome assemblies**

| | | Estimated time since divergence (MYA) | | |
| --- | --- | --- | --- | --- |
| | | R. catesbeiana | N. parkeri | X. tropicalis |
| Estimated identity (%) | R. catesbeiana | – | 89.0 | 208.6 |
| | N. parkeri | $86.01 \pm 3.32 \times 10^{-4}$ | – | 208.6 |
| | X. tropicalis | $79.16 \pm 3.42 \times 10^{-3}$ | $77.92 \pm 2.51 \times 10^{-3}$ | – |

k = 25, Sequences ≥ 500 bp
Note that *R. catesbeiana* and *N. parkeri* are hypothesized to share a common ancestor that diverged from the ancestor of *X. tropicalis*; this is reflected in the identical estimated time since divergence from the *X. tropicalis* ancestor for these two organisms

phylogenetic trees. At least at the whole Mt genomic level, the *Lithobates* group branches out of the *Rana* group, as opposed to forming a distinct clade such as what is observed for salamanders and toads.

Typically individual Mt genes have been used for phylogenetic analyses. Comparing the specific Mt gene *cyb*, *Rana* and *Lithobates* often branch together indicative of the close genetic conservation of these species, but do not form independent clades, which suggests a high degree of sequence conservation instead of the divergence observed between distinct genera (Fig. 2b; Supplementary Fig. 6). Ribosomal RNA genes *rnr1* and *rnr2* show phylogenies similar to that of the entire Mt genome, this time with *Rana* branching out of the *Lithobates* clade (Fig. 2c, d; Supplementary Figs. 7, 8).

**Comparative genomics**. Three diploid frog genomes have now been sequenced, so the degree of sequence identity over the whole genomes of *N. parkeri*, *X. tropicalis*, and *R. catesbeiana* may be estimated. We performed this analysis using Bloom filters[29], which are probabilistic data structures with bit sets for each genome's k-mers (collection of all subsequences of length k). In a previous study, this method was shown to provide concordant estimates of the genome sequence divergence of known model organisms (human and apes), and was applied to conifer genomes[30].

This method is designed to compare the k-mer spectra of any two genomes by computing the k-mer set bit intersections of their respective Bloom filters. It is assumed that differences between the genomes are independently distributed. We point out that this method does not factor size differences in the genomes, nor structural rearrangements; instead it reports on the commonality over short sequence stretches. We also note that the method is not applicable to very divergent genomes, where common k-mers are rarer, and precise values of sequence identity are not expected. As such, genome-scale divergence figures are likely an underestimate of their true separation.

At k = 25 bp, *R. catesbeiana* shares higher sequence identity (mean ± SD, $86.0 \pm 3.3 \times 10^{-4}$ %) with the High Himalaya frog (*N. parkeri*) than to the more evolutionarily diverged *X. tropicalis* ($79.2 \pm 3.4 \times 10^{-4}$ %), which is estimated to have shared a common ancestor with bullfrog >200 MYA, more than twice as much (89 MYA) as between *N. parkeri* and *R. catesbeiana* (Table 2).

To put the *N. parkeri* and *R. catesbeiana* values into context, we compared three mammalian species predicted to have diverged within the same time scale[12]. The sequence identity between *Homo sapiens* and *Rattus norvegicus* (Norway rat) or *Oryctolagus cuniculus* (European rabbit), was also estimated using the same settings as for *Rana*, *Nanorana*, and *Xenopus*. *H. sapiens* has a lower sequence identity with *R. norvegicus* ($81.0 \pm 2.4 \times 10^{-3}$ %) than with *O. cuniculus* ($83.1 \pm 4.4 \times 10^{-4}$ %) (Supplementary Table 10).

## Discussion

Amphibians are the only group where most of its members exhibit a life cycle that includes distinct independent aquatic larval and terrestrial juvenile/adult phases. The transition between the larval and juvenile phases requires substantial or complete remodeling of the organism (metamorphosis) in anticipation of a terrestrial lifestyle. Thus, this places amphibians in a unique position for the assessment of toxicological effects in both aquatic and terrestrial environments. As a model for human and mammalian perinatal development, including the transition from the aquatic environment of the womb to the outside world, *R. catesbeiana* is a preferable model to the *Xenopus* species, because of similar physiological transformations. In contrast, *Xenopus* remain aquatic throughout life[4].

Our sequence divergence analysis with the recently released *Xenopus* genome of a diploid species, *X. tropicalis* (version 9.0) highlights the considerable evolutionary divergence between the Pipids and Ranids, and underscores the need for Ranid-specific genomic resources. Our present work is consistent with earlier estimates of divergence dating over ~200 MYA. The initial assembly of another *Xenopus* species' genome – that of the allotetraploid *X. laevis* inbred 'J' strain—has been published recently[6]. The haploid genomes of these species are substantially smaller (1.7 Gbp and 3.1 Gbp, respectively) than the typical Ranid genome (www.genomesize.com). Despite the difference in relative sizes, the number of high confidence predicted protein-encoding genes in *R. catesbeiana* (~22,000) is comparable (24,022 in *X. laevis*; ~20,000 in *X. tropicalis*)[6,31]. However, the sequence divergence at the nucleic acid level confirms the empirical challenges of using *Xenopus* genomes as scaffolds for RNA-Seq experiments in Ranids. Even the recently published *N. parkeri* genome[11] is substantially separated from *R. catesbeiana* by nearly 90 million years of evolutionary time[12]. It is interesting to note that while this degree of evolutionary separation is similar to that of humans from rats and rabbits, these mammals have a greater degree of sequence divergence than *N. parkeri* and *R. catesbeiana*. Unfortunately, the paucity of amphibian genomes precludes assessment as to how general this property may be within the Neobatrachia suborder to which both species belong.

The taxonomic classification of Ranid species is contentious and highly debated amongst scholars[27]. This stems largely from the suggested reclassification of the genus *Rana* in favor of *Lithobates* a few years ago[28]. Our phylogenetic analyses based on comparisons at the nucleotide level of complete mitochondrial genomes and genes from selected salamanders, toads, and Ranids does not offer a rationale for the proposed change. It instead supports a close relationship between species classified as *Lithobates*, which may be considered a subgenus within the *Rana* genus. This observation is consistent with the recent phylogenetic analysis by Yuan et al.[32].

The molecular mechanisms of amphibian metamorphosis have been predominantly studied using *X. laevis* and *X. tropicalis*,

**Table 3 Sequencing data for *Rana catesbeiana* genome assembly**

| Library protocol | Read length (bp) | Sequencing platform | Nominal fragment length (bp) | # Libraries | # Reads (M) | Fold coverage |
|---|---|---|---|---|---|---|
| PET | 150 | HiSeq 2000 | 600 | 8 | 1,187 | 30 |
| PET | 250 | HiSeq 2500 | 550 | 4 | 736 | 31 |
| PET | 500 | MiSeq | 600 | 8 (same) | 66 | 5 |
| MPET | 100 | HiSeq 2000 | 9,000–13,000 | 4 | 262 | N/A |
| SLR | N/A | HiSeq 2000 | 500–14,000 | 1 | 0.6 | 0.5 |
| 10XG | 150 | HiSeq X | 400 | 3 | 1,625 | N/A |

likely in no small part because of their amenability to captive breeding, which ensures a ready supply of research specimens. However, *Xenopus* larvae are typically much smaller than those of *R. catesbeiana*, with the consequence that each individual animal yields a smaller quantity of tissue for analysis. Indeed, *R. catesbeiana* tadpoles are large enough that techniques such as the cultured tail fin (C-fin) assay are possible, where multiple tail fin tissue biopsies are collected from an individual animal and cultured ex vivo in a variety of hormone or chemical conditions[33]. With this assay design each animal is effectively exposed simultaneously and independently to every condition in the experiment, and the result of these different conditions can be evaluated within each individual animal using powerful repeated-measures statistics. Similar assays are possible using the back skin[34] and lungs[35]. These types of approaches support the objective of reducing and refining animal studies while retaining biological relevance to the intact organism.

Analysis of TH-induced changes in tadpole back skin gene expression revealed an unprecedented view of the activation of new gene expression programs as an integral part of the transition from larval to adult skin. This process involves apoptosis of the terminal larval skin cells, proliferation of progenitor cells, and their differentiation into adult skin cells[36]. It is notable that the largest set of differentially expressed transcripts is involved in transcription and RNA/DNA processing roles. The expanding population of progenitor cells, which is enriched in the T3-treated samples relative to the samples from the control animal skin, drives differential expression of genes with DNA replication, histone mRNA metabolism, and RNA processing functions. As the level of circulating TH increases in the tadpole, expression of key cell cycle control genes changes to regulate the proliferation of skin cells, including cyclin C and cyclin B[37–39]. This is in contrast to TH-induced remodeling of the liver tissue, where these cellular processes were not as prominently represented[40].

Current lncRNA databases are mostly populated with sequences that were derived from human and mouse. In addition to expanding the effectiveness of RNA-Seq analyses through annotation of protein-encoding transcripts, the present study identified over 6,000 putative lncRNA candidates in the bullfrog. Despite being non-coding with relatively low level of sequence conservation, some lncRNAs contribute to structural organization, function, and evolution of genomes[41]. Examples include classical lncRNAs, such as X-inactive specific transcript (XIST), HOX transcript antisense RNA (HOTAIR), telomerase RNA component (TERC), and many more with roles in imprinting genomic loci, transcription, splicing, translation, nuclear factor trafficking, chromatin modification, shaping chromosome conformation, and allosterically regulating enzymatic activity (reviewed in Geisler and Coller[42]). These functional roles overlap with the major gene ontologies associated with the protein coding genes differentially expressed in the T3-treated back skin, and it is reasonable to conclude that some of the candidate lncRNAs that we identified participate in these critical biological processes.

Dynamic regulation of lncRNAs has previously been observed during embryogenesis in *X. tropicalis*[43]. Additional studies have ascribed roles of lncRNAs in differentiation of mouse embryonic stem cells[44], and shown tissue specific patterns of expression in human tissues[45]. HOTAIR expression is transcriptionally induced by estradiol in the MCF7 breast cancer cell line, and its promoter contains multiple estrogen response elements[46]. As the candidate lncRNAs that we identified represented 1/6th of the differentially expressed genes in response to TH treatment, this suggests an important role for lncRNA in the amphibian metamorphic gene expression program initiated by this hormone. The data presented herein extend hormonal regulation of lncRNAs to postembryonic developmental processes.

The bullfrog genome is larger than those of *N. parkeri*, *X. laevis*, and *X. tropicalis*, in concordance with earlier predictions for many Ranid genomes[6,7,11,47]. Unlike *X. laevis*, this does not appear to be due to an allotetraploidization event in the Ranid progenitor species[48]. Another possibility for genome enlargement is integration and propagation of foreign DNA, which can manifest as DNA sequence repeat elements. Many of these integrations are likely to be derived from ancestral integration of viral genomes into the host genome. The higher proportion of repetitive DNA in the bullfrog genome relative to *N. parkeri* and *Xenopus* is likely responsible for much of the genome enlargement in Ranids.

The *R. catesbeiana* genome presented herein provides an unprecedented resource for Ranidae. For example, it will inform the design and/or interpretation of high throughput transcriptome sequencing (RNA-Seq), chromatin immunoprecipitation sequencing (ChIP-Seq), and proteomics experiments. We anticipate that this resource will be valuable for conservation efforts such as identifying host/pathogen interactions and to identify environmental impacts of climate change and pollution on the development and reproduction of Ranid species worldwide.

## Methods

**Sample collection**. Liver tissue was collected from an adult male *R. catesbeiana* specimen that was caught in Victoria, BC, Canada and housed at the University of Victoria Outdoor Aquatics Unit. The tissue was taken under the appropriate sanctioned protocols and permits approved by the University of Victoria Animal Care Committee (Protocol #2015-028) and British Columbia Ministry of Forests, Lands and Natural Resource Operations (MFLNRO) permit VI11-71459. This frog was euthanized using 1% w/v tricaine methane sulfonate in dechlorinated municipal water containing 25 mM sodium bicarbonate before tissue collection. Dissected liver pieces were preserved in RNA*later* (Thermo Fisher Scientific Inc., Waltham, MA, USA) at room temperature followed by incubation at 4 °C for 24 h. Tissue samples were subsequently moved to storage at −20 °C before DNA isolation. Total DNA was isolated using the DNeasy Blood and Tissue Kit (QIAGEN Inc., Mississauga, ON, Canada; Cat# 69506) with the inclusion of RNase treatment as per the manufacturer's protocol, and stored at −20 °C before library preparation.

**DNA library preparation and sequencing**. All reagent kits used were from the same vendor (Illumina, San Diego, CA) unless otherwise stated. Two sets of PET libraries were constructed: (1) 16 libraries were produced using 1 μg of DNA using custom NEBNext DNA Library Prep Reagents from New England BioLabs Canada (Whitby, ON); and (2) four libraries were constructed using 0.5 μg of DNA and the custom NEB Paired-End Sample Prep Premix Kit (New England BioLabs Canada). DNA sequence reads were generated from these libraries according to the manufacturer's instructions on the Illumina HiSeq 2000 platform (Illumina, San Diego, CA) in "High Throughput" mode with the HiSeq SBS Kit v3, on the Illumina HiSeq

2500 platform in "Rapid" mode with the HiSeq Rapid SBS kit v1, or on the Illumina MiSeq platform with the MiSeq Reagent Kit v2. See Table 3 for additional details.

The MPET (a.k.a. jumping) libraries were constructed using 4 µg of DNA and the Nextera Mate Pair Library Preparation Kit, according to the manufacturer's protocol, and 100 bp paired-end reads were generated on the Illumina HiSeq 2000 platform with the HiSeq SBS Kit v3. The Synthetic Long-Read (SLR, a.k.a. Moleculo) library was constructed using 500 ng DNA and Illumina's TruSeq SLR library prep kit with 8–10 kb size DNA fragments. Libraries were loaded on an Illumina HiSeq 2500 platform for 125 bp paired end sequencing.

High molecular weight DNA for 10x Genomics Chromium linked reads was prepared using a MagAttract HMW DNA kit (QIAGEN Cat# 67563). Integrity of the DNA was checked using Pulsed-field gel electrophoresis (PFGE). Using the 10x Genomics Chromium Controller instrument (10x Genomics, Pleasanton, CA) fitted with a micro-fluidic Genome Chip, for each replicate, a library of Genome Gel Beads was combined with 1 ng of gDNA, Master Mix and partitioning oil to create Gel Bead-In-EMulsions (GEMs). The GEMs were subjected to an isothermal amplification step and barcoded DNA fragments through Illumina library construction according to the Chromium Genome Reagent Kits Version 2 User Guide. qPCR was performed to assess library yield and an Agilent 2100 Bioanalyzer DNA 1000 chip was run to determine the library size range and distribution. Two libraries were pooled and loaded in one lane on an Illumina HiSeq X sequencer, while the third library was loaded in another lane of the same flowcell, and 150 bp paired end reads were generated.

Combined, this approach accounted for 66-fold sequence coverage of the ~6 Gbp bullfrog genome (Table 3).

**Computing hardware.** Sequence assemblies were performed on high performance computer clusters located at the Canada's Michael Smith Genome Sciences Centre, and consisted of nodes with 48 GB of RAM and dual Intel Xeon X-5650 2.66 GHz CPUs running Linux CentOS 5.4 or 128 GB of RAM and dual Intel Xeon E5-2650 2.6 GHz CPUs running Linux CentOS 6. Computational analyses used either this hardware, or nodes consisting of 24 GB of RAM and dual Intel Xeon X-5550 2.67 GHz CPUs running Red Hat Enterprise Linux 5 as part of WestGrid, Compute Canada.

**Read merging.** PET read pairs were merged sequentially using the ABySS-mergepairs tool[49] and Konnector (version 1.9.0)[50]. Bloom filters were constructed from all reads using the ABySS-Bloom utility[30], and every tenth value of k between 75 and 245 bp, inclusive. Reads from potentially mixed clusters on the sequencing flow cells (determined by the Illumina chastity flag) were discarded, and the remaining reads were trimmed to the first base above a quality threshold (Q = 3 on the phred scale) before merging.

**Assembly process.** ABySS (version 1.9.0) was used to reconstruct the *R. catesbeiana* genome[51]. For the initial sequence assembly, three sets of reads were used: (i) merged reads described above from paired-end Illumina HiSeq 150 bp, 250 bp, and MiSeq 500 bp libraries, (ii) unmerged reads from these same libraries, and (iii) synthetic long-reads. The unmerged HiSeq and MiSeq PET reads were also used for paired linking information to generate contigs. Finally, the MPET reads were used to bridge over regions of repetitive sequence to form scaffolds (see Table 3 for summary statistics of the sequencing data).

Automated recovery of unresolved bases within these scaffolds was performed using Sealer[18] version 1.9.0 (k = 245–75:10). Sealer uses a Bloom filter representation of a de Bruijn graph constructed using k-mers derived from the genomic reads to find a path between the sequences flanking scaffold gaps, and fill in the consensus sequence. In comparison to the fixed k-mer length of the whole genome assembly method, it uses a range of k-mer lengths to navigate repeat and low coverage areas within the graph. The Bloom filters that were used during the read merging phase were reused, and default values were used for all parameters except for "–flank-length = 260" and "–max-paths = 10".

The resulting ABySS scaffold assembly (k = 160 bp) was rescaffolded with RAILS version 0.1[15] (-d 250 -i 0.99) using both SLR data and Kollector[14] targeted gene reconstructions (TGA; Supplementary Table 1 and Supplementary Methods). In RAILS, long sequences are aligned against a draft assembly (BWA-MEM v0.7.13-r1126, -a -t16)[52], and the alignments are parsed and inspected, tracking the position and orientation of each in assembly draft sequences, satisfying minimum alignment requirements (a minimum of 250 anchoring bases with 99% sequence identity or more was used in this study). Sequence scaffolding is performed using the scaffolder algorithms from LINKS[16], modified to automatically fill gaps with the sequence that informed the merge. The resulting assembly was sequentially rescaffolded with a composite reference transcriptome (Bullfrog Annotation Resource for the Transcriptome; BART, see below and Supplementary Table 1) with ABySS-longseqdist v1.9.0 (l = 50, S = 1000:-)[30]. It was further rescaffolded iteratively with LINKS (v1.7) using a variety of long sequence data (Supplementary Table 1) including SLR data (10 iterations –d 1–10:1 kbp, –t 10–1:-1, –k 20), MPET (-k 20, -t 5, -d 7.1 kbp, -e 0.9) and other assembly draft data (Kollector targeted reconstructions and whole genome assembly at k = 128 bp combined, 7 iterations –d 1–15:2.5 kbp, –t 20,10,5,5,4,4,4 k = 20). These scaffolds were subjected to automated gap closing with Sealer to form the version 2 assembly. A final round of scaffolding using 10x Genomics Chromium linked reads and ARCS v1.0.1 (−s

80 −c 5 −l 0 −d 0 −r 0.05 −e 30000 −v 1 −m 20–4000 −z 500 and LINKS v1.8.5 −l 2 −a 0.9)[17], followed by application of Sealer to close gaps, yielded the version 3 assembly. See Supplementary Methods for additional description of the assembly versions. Completeness of the assemblies was evaluated by comparison to a set of ultra-conserved core eukaryotic genes[20] and near-universal single-copy orthologs[13].

**Protein coding gene prediction.** The MAKER2 genome annotation pipeline (version 2.31.8) was used to predict genes in the draft *R. catesbeiana* genome[19] (see Supplementary Methods for additional details, including repetitive sequence element detection). We refined the MAKER2 predicted gene list further by identifying a high confidence set, better suited for downstream biological analyses. Three criteria were considered in the generation of the high confidence gene set with a gene being added when it satisfied one or more of the following criteria: (1) the predicted transcripts must have at least one splice site, and all putative splice sites must be confirmed by an alignment to external transcript evidence; (2) the coding DNA sequence (CDS) of each transcript must have a BLASTn[53] alignment to a BART contig with at least 95% identity along 99% of its length; or (3) the protein sequence encoded by the CDS must have a BLASTp[53] alignment to a human or amphibian Swiss-Prot protein sequence[54] (retrieved 16 February 2016) with at least 50% identity along 90% of its length (Supplementary Fig. 1).

**Functional annotation.** The high confidence set of transcripts was annotated according to the best BLASTp alignment of each putative encoded protein to the Swiss-Prot database[54], provided that they aligned with at least 25% identity along 50% of their length. Proteins that did not have a sufficient SwissProt hit but did have a significant HMMER[55] (version 3.1b2) alignment to a Pfam[21] (release 29.0) model (E-value < 0.05) were assigned a putative domain-based annotation. There were two levels of confidence for the annotations: (1) the most robust were identified using GeneValidator[22], which compares protein coding gene predictions with similar database proteins, where those having a score of 90 or greater were definitively identified as the Swiss-Prot sequence they aligned to; and (2) all other predicted transcripts were considered to encode "hypothetical" proteins. These hypothetical proteins were labeled as 'similar to' their Swiss-Prot hit, or as a containing a particular Pfam domain or domains.

**Construction of a composite reference transcriptome.** Transcriptome assemblies were generated from 32 *R. catesbeiana* tadpole samples (representing 5 tissues under several different chemical and temperature exposure conditions) using Trans-ABySS[56] (Supplementary Table 5). The transcripts from each independent assembly were aligned using the BLAST-like Alignment Tool[57] or parallelized BLAT (pblat; icebert.github.io/pblat/) to identify highly similar sequences, where only the longest example of each set of similar sequences was retained. This process produced 794,291 transcripts 500 bp or longer, resulting in a composite reference transcriptome (Bullfrog Annotation Resource for the Transcriptome; BART). Further, we report 1,341,707 transcripts between 200 and 499 bp long (termed BART Jr.).

**lncRNA prediction.** To complement the protein coding gene predictions, a computational pipeline was developed to identify putative lncRNAs in the *R. catesbeiana* composite reference transcriptome BART. As there is a paucity of conserved sequence features that may positively identify lncRNA transcripts, we instead took a subtractive approach, and omitted transcripts that were predicted to have coding potential or had sequence similarity to known protein encoding transcripts, as has been advocated in previous studies[58]. See Supplementary Methods for additional details.

We then used CD-HIT-EST[59] (v4.6.6, -c 0.99) to identify and remove contigs with significantly redundant sequence content. The remaining transcripts were then interrogated for the presence of a poly(A) tail and one of 16 polyadenylation signal hexamer motifs (see Supplementary Table 6). The contigs were aligned to the genome assembly using GMAP (v2016-05-01, -f samse, -t 20)[60], and instances where there was a 3′ sequence mismatch due to a run of As, or a 5′ mismatch due to a run of Ts (in cases where the strand specific sequencing failed, and an RNA molecule complementary to the actual transcript was sequenced) prompted a search for the presence of a hexamer motif within 50 bp upstream (relative to the direction of coding) of the putative transcript cleavage site. Contigs containing a poly(A) tail and a hexamer motif were selected for further analysis. We are aware that not all lncRNA are polyadenylated. The poly(A) tail filter was put in place to reduce the proportion of spurious transcripts, retained introns and assembly artifacts.

Candidate lncRNA transcripts were aligned to the draft genome with GMAP[60] (version 2015-12-31, -f 2, -n 2, –suboptimal-score = 0, –min-trimmed-coverage = 0.9, –min-identity = 0.9), and those that had at least 90% of their sequence identified across no more than two separate genomic scaffolds with 90% sequence identity were retained. Alignments where the exon arrangement was not collinear with the original contig sequence were omitted. Further evidence of conservation of lncRNA candidates among amphibian species was obtained using a comprehensive amphibian transcriptome shotgun assembly database, as described in the Supplementary Methods and Supplementary Table 7.

**Differential gene expression analysis.** As an example of the utility of the draft genome assembly and high confidence gene predictions, RNA-Seq reads from six premetamorphic *R. catesbeiana* tadpoles exposed to 10 pmol/g body weight T3 or dilute sodium hydroxide vehicle control for 48 h were used to characterize the T3-induced gene expression program in the back skin (Supplementary Methods; Supplementary Table 5). Experimental protocols were approved by the University of Victoria Animal Care Committee (Protocol #2015-028). The 100 bp paired-end reads were aligned to the version 2 draft genome using STAR[61] (version 2.4.1d, –alignIntronMin 30, –alignIntronMax 500000, –outFilterIntronMotifs Remove-NoncanonicalUnannotated, –outFilterMismatchNMax 10, –sjdbOverhang 100), and read counts per transcript were quantified using HTSeq[62] (version 0.6.1, default settings). Differential expression in response to T3 treatment was assessed using the DESeq2 software package[63] (version 1.10.1, alpha = 0.05), and significance was considered where the Benjamini—Hochberg adjusted p-value was less than 0.05. Transcripts with zero counts in all six samples were excluded from the analysis. qPCR analysis of transcripts and gene ontology analysis is described in the Supplementary Methods.

**Mitochondrial genome assembly and finishing.** The mitochondrial (Mt) genome sequence was identified integrally in our whole genome assembly. Rounds of scaffolding effectively brought unincorporated Mt contigs to the edge of the scaffold, and after inspection were removed by breaking the redundant scaffolds at N's. Multiple sequence alignments were done between our sequence, two NCBI references originating from China (GenBank accessions NC02296 and KF049927.1) and one from Japan (AB761267), using MUSCLE[64] from the MEGA phylogenetic analysis package[65] using default values. These analyses indicated that the Mt sequence reported herein is most similar to the Japanese sequence, but with some discrepancies in two repeat regions, namely the absence of a 161 bp sequence at coordinate 15,270, and a 12-bp insertion at coordinate 9214 relative to AB761267. We resolved these misassemblies using the correct Japanese reference sequence for these regions as candidates for a targeted de novo assembly of Illumina paired-end 250 bp reads with TASR[66]; v1.7 with -w 1 -i 1 -k 15 -m 20 -o 2 -r 0.7. TASR uses whole reads for assemblies, and mitigates misassemblies otherwise caused by breaking reads into k-mers. The resulting TASR contigs that captured the correct sequences were inserted into our assembly in the corresponding regions. Transfer RNA (tRNA) and protein coding genes were annotated by GMAP alignment of the gene sequences included in KF049927.1.

**Phylogenetic analyses.** Complete mitochondrial genome sequences of selected salamanders and frogs (Supplementary Table 8) were compared using the MEGA phylogenetic package[65]. In another set of phylogenies, we also compared the mitochondrial genes *cyb* and 12 s and 16 s rRNA *rnr1* and *rnr2* of selected amphibian species (Supplementary Table 9). For these analyses, we first generated multiple sequence alignments of the genome or gene sequences described above using either MUSCLE[64] or clustalw[67] (v1.83 with gap opening and extension penalty of 15 and 6.66 for both the pairwise and multiple alignment stages, DNA weight matrix IUB with transition weight of 0.5), and used the resulting pairwise alignments as input for MEGA7[65]. The evolutionary history was inferred by using the Maximum Likelihood method based on the Tamura-Nei model[68], where initial trees for the heuristic search are obtained by applying Neighbor-Join and BioNJ algorithms to a matrix of pairwise distances estimated using the Maximum Composite Likelihood approach, and then selecting the topology with superior log likelihood value.

**Comparative genome analysis using Bloom filters.** The genomes of *N. parkeri* (version 2; http://dx.doi.org/10.5524/100132), *X. tropicalis* (version 9.0, downloaded from xenbase.org) and *R. catesbeiana* (version 2; the present study) were compared for their k-mer (k = 25) contents using ABySS-Bloom, a utility designed to approximate sequence divergence between draft genomes[30]. In addition, the *H. sapiens* genome (GRCh38, downloaded from NCBI) was compared to the *O. cuniculus* (version 2.0, downloaded from Ensembl) and *R. norvegicus* (version 6.0, downloaded from Ensembl) genomes using the same method. The latter two species are both estimated to be separated from *H. sapiens* by ~90 million years of evolutionary time[12] (Supplementary Table 10), which is nearly the same as the estimate for separation of *R. catesbeiana* from *N. parkeri*.

**Data availability.** The whole genome sequence data and assembly versions 2 and 3 of the North American bullfrog genome with annotated MAKER2 gene predictions are available at NCBI Genbank under accession LIAG00000000, BioProject PRJNA285814.

The Mt genome assembly was submitted to NCBI GenBank under accession KX686108. The RNA-Seq reads and assembled BART contigs are available under NCBI BioProject PRJNA286013, and the BART Jr. contigs (collection of short transcripts between 200 and 499 bp in length in the composite transcriptome that forms the basis for BART) are available from the BCGSC ftp site: ftp.bcgsc.ca/supplementary/bullfrog. The candidate lncRNA sequences and genomic coordinates are also available from the BCGSC ftp site.

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

## Acknowledgements

This work has been supported by Genome British Columbia and Natural Sciences and Engineering Research Council of Canada, with additional support provided by the National Human Genome Research Institute of the National Institutes of Health (under award number R01HG007182). The content is solely the responsibility of the authors, and does not necessarily represent the official views of the National Institutes of Health or other funding organizations. We thank Dr. Belaid Moa for advanced research computing support from WestGrid, Compute Canada, and the University of Victoria computing systems.

## Author contributions

C.C.H. and I.B. conceived the study. The genomic DNA was prepared by N.V. and S.A.H. Sequencing libraries and reads were generated by P.P., He.K., Y.J., M.J., A.J.M., R.C., S.P., R.A.M., and R.A.H. The draft nuclear genome was assembled and scaffolded by S.A.H., and R.L.W., B.P.V.. S.A.H. generated the gene predictions and annotations, and constructed the BART reference transcriptome. R.L.W. assembled the Mt genome and performed the phylogenetic analyses. B.P.V. performed the Bloom filter comparative genome analysis. The candidate lncRNA sequences were identified by Ha.K. and E.A.G. The differential gene expression analysis was performed by S.A.H. Quantitative PCR primers were designed and applied by N.V, J.M.R., S.O., and B.V.W. The Kollector TGA sequences were generated by E.K. S.A.H., R.L.W., C.C.H., and I.B. prepared the manuscript with contributions from E.K., Ha.K., and Y.J.

## Additional information

**Competing interests:** The authors declare no competing financial interests.

