## [Peer Review File · Nature Communications]

Reviewers' comments:

Reviewer #1 (Remarks to the Author):

The Manuscript "Draft genome of the North American bullfrog (*Rana* [*Lithobates*] *catesbeiana*)" provides a description of the initial assembly and analysis of the bullfrog genome. The species fills an important phylogenetic gap within the frogs and it appears that most of the analyses related to assembly and annotation are robust, however the description of the results seems to undersell (at times) the underlying analyses or appears to lack focus in presenting the data. I include a few comments below that (when addressed) will improve the interpretability of the manuscript.

Primary Comments:

- 1) In a few cases the authors seem to pitch the *Xenopus* assemblies in a negative light: "are of limited utility" , "annotated to some extent" ... I think some careful editing to avoid this would make the manuscript would improve the receptivity of the manuscript to that community (which does not include this reviewer).
- 2) The sentence on Page 4, line 68-70 is difficult to read and seems to lean toward grandiose. I would consider careful revision.
- 3) lncRNA Results – these seem to be briefer and vaguer than necessary. I would include more detail that relays the rigor of those analyses and hint at the potential biological significance (including differential expression).
- 4) Figure 1 – It is not clear why median count was used as versus mean. It might also be useful to incorporate some measure of statistical significance (similar to a volcano plot).
- 5) Phylogeny Results – A controversy is mentioned in the introduction but now well explained.
- 6) Figure C needs taxon labels or something similar for all branches in panels B-D.
- 7) Table 3 – It is unclear why percent identities are different for *R. catesbeiana*/*X. tropicalis* and *N. parkeri*/*X. tropicalis*, yet the estimated divergence dates are identical.
- 8) The authors describe the importance of the c-fin assay, yet they present data from back skin. It is unclear why.
- 9) Page 17 line 335 – "A certain fraction" could be stated more precisely (e.g. 1.5%).
- 10) Page 19 – The paragraph describing section of this high confidence subset is a bit confusing. I think it should be introduced early in the paragraph that the genes must fit at least one of the three criteria.

Minor Comments:

- 1) in a few instances, definitions follow abbreviations (PET, BART) or appear to be unnecessary (c-fin)
- 2) Line 155 – Should "genetic" be "genic"

Reviewer #2 (Remarks to the Author):

In their manuscript, "Draft genome of the North American bullfrog (*Rana* [Lithobates] catesbeiana), Hammond et al. describe the sequencing, assembly, annotation and analysis of the North American bullfrog. The bullfrog genome was a daunting task as a reference genome project – a 6 Gb genome is a more challenging assembly than is faced by most vertebrate assembly teams. This goes some way to explaining why the assembly described here is so fragmented – with only a contig N50 of 5kb and a scaffold N50 of 44kb. Even though the poor quality of the assembly is understandable, it remains that an assembly this fragmented will yield a poor gene annotation (as noted by the authors, only 41% of the CEGMA genes were complete, and only 86% were found even partially), and will be therefore of limited use to the greater genomics community. It also reduces the power of the gene-based analyses found in this manuscript.

As for the analyses, I think that in general, there needs to be more description of the rationale and significance of the experiments, especially in regard the differential expression analysis and the non-iridovirus sequence. Overall, it was often difficult to understand what conclusions the authors drew from the work described here. Specifically, with regard to the differential expression experiment, the authors speculated that RNA/DNA processing genes predominated because transcripts involved in thyroid hormone-dependent processes require capping and polyA-adenylation. But since all transcripts require that processing, I'm not sure what relevance that explanation has. I was also confused about the amount of description of lncRNAs and their importance to biology since the only relevant data in this paper was a total number of lncRNAs annotated in this assembly. In addition, I wasn't sure why the authors didn't use a larger dataset than mtDNA to make an amphibian phylogeny since they have so much more sequence they could use. There are salamander chromosomes that have been sequenced, I believe, that could be used for this purpose. And once again, I do not know what conclusions the authors drew from the estimated sequence identity between frogs and toads.

Smaller points:

- Please don't cite timetree.org for divergence times, instead cite the papers that timetree cites
- I love that the authors named their short transcript transcriptome 'BART Jr.'

Reviewers' comments:

Reviewer #1 (Remarks to the Author):

The Manuscript “Draft genome of the North American bullfrog (Rana [Lithobates] catesbeiana)” provides a description of the initial assembly and analysis of the bullfrog genome. The species fills an important phylogenetic gap within the frogs and it appears that most of the analyses related to assembly and annotation are robust, however the description of the results seems to undersell (at times) the underlying analyses or appears to lack focus in presenting the data. I include a few comments below that (when addressed) will improve the interpretability of the manuscript.

We thank the Reviewer for the comments and suggestions. We have addressed each point below and in the main manuscript text.

Primary Comments:

P1. In a few cases the authors seem to pitch the *Xenopus* assemblies in a negative light: “are of limited utility”, “annotated to some extent” ... I think some careful editing to avoid this would make the manuscript would improve the receptivity of the manuscript to that community (which does not include this reviewer).

R1. Our discussion of the *Xenopus* data has been recast to note the data gaps in amphibian genomics, rather than specifically identify limitations of the previously available data. We agree that this is a better approach, and we thank the Reviewer for their insight.

P2. The sentence on Page 4, line 68-70 is difficult to read and seems to lean toward grandiose. I would consider careful revision.

R2. While we feel that our work will be highly impactful in the fields we list, we do acknowledge that our bullfrog genome represents an initial draft. With that in mind, we have tempered our language in this section and in the rest of the manuscript.

P3. lncRNA Results – these seem to be briefer and vaguer than necessary. I would include more detail that relays the rigor of those analyses and hint at the potential biological significance (including differential expression).

R3. The text has been updated, giving additional details of our lncRNA detection approach. In the results section, we place more emphasis on the proportion of differentially expressed candidate lncRNA transcripts and discuss further the possible implications these may have on frog biology.

P4. Figure 1 – It is not clear why median count was used as versus mean. It might also be useful to incorporate some measure of statistical significance (similar to a volcano plot).

R4. Figure 1 depicts the median expression level per transcript for the three T3-treated animals and for the three control animals. This representation of the average DESeq2-normalized abundance is purely for graphical purposes, as DESeq2 performs its own sample weighting and normalization of the raw counts per gene prior to fitting a negative binomial general linear model. We considered significance at a Benjamini-Hochberg

adjusted p-value of 0.05; these genes are coloured pink rather than black. These details have been added to the figure legend.

P5. Phylogeny Results – A controversy is mentioned in the introduction but now well explained.

R5. We have modified the main manuscript text to clarify the main point of contention.

P6. Figure C needs taxon labels or something similar for all branches in panels B-D.

R6. These labels have been added to panels B-D, and we agree that they aid reader understanding.

P7. Table 3 – It is unclear why percent identities are different for *R. catesbeiana*/*X. tropicalis* and *N. parkeri*/*X. tropicalis*, yet the estimated divergence dates are identical.

R7. The estimated divergence date represents the time to the most recent common ancestor. *R. catesbeiana* and *N. parkeri* are both of the Anuran suborder *Neobatrachia*, which is hypothesized to descend from an ancestor that diverged from that of *X. laevis*, of the suborder *Mesobatrachia*. Hence, both *R. catesbeiana* and *N. parkeri* are expected to have the same divergence date from *X. laevis*. We clarify this important point in the Table 3 caption.

P8. The authors describe the importance of the c-fin assay, yet they present data from back skin. It is unclear why.

R8. Our description of the different assays made possible in the bullfrog due to its larger size relative to the *Xenopus* species should have included reference to assays using the back skin and lungs. We have clarified the manuscript text on page 13.

P9. Page 17 line 335 – “A certain fraction” could be stated more precisely (e.g. 1.5%).

R9. We have altered this sentence to note only that Sealer was used for automated recovery of unresolved bases, and limit reporting the success of this operation to the Results section.

P10. Page 19 – The paragraph describing section of this high confidence subset is a bit confusing. I think it should be introduced early in the paragraph that the genes must fit at least one of the three criteria.

R10. We have updated the paragraph on p. 21 to note that “a gene that satisfied one or more of these criteria was added to the high confidence set”.

Minor Comments:

P11. In a few instances, definitions follow abbreviations (PET, BART) or appear to be unnecessary (c-fin)

R11. We thank the Reviewer for their careful reading. We have corrected out-of-order definitions and abbreviations. The C-fin abbreviation remains because it is the defined

short form name of that assay and readers may be more (or only) familiar with it.

P12. Line 155 – Should “genetic” be “genic”

R12. That sentence has been clarified to refer to the whole mitochondrial genome (Panel A), rather than the individual mitochondrial genes in the other panels.

Reviewer #2 (Remarks to the Author):

P1. The bullfrog genome was a daunting task as a reference genome project – a 6 Gb genome is a more challenging assembly than is faced by most vertebrate assembly teams. This goes some way to explaining why the assembly described here is so fragmented – with only a contig N50 of 5kb and a scaffold N50 of 44kb. Even though the poor quality of the assembly is understandable, it remains that an assembly this fragmented will yield a poor gene annotation (as noted by the authors, only 41% of the CEGMA genes were complete, and only 86% were found even partially), and will be therefore of limited use to the greater genomics community. It also reduces the power of the gene-based analyses found in this manuscript.

R1. We thank the Reviewer for recognizing the difficulty in assembling the bullfrog genome. Since submitting our manuscript, we have improved our assembly through the use of Chromium linked reads from 10X Genomics. Using the ARCS scaffolding software recently developed in our group, the scaffold NG50 length of the draft bullfrog genome is now 69 kbp; an increase of over 11 kbp. As well, the N50 of the scaffolds' constituent

contigs, also called scaftigs, now stands at 6.7 kbp.

The improved quality of our assembly is also reflected in the genome completeness measures provided by BUSCO, the successor to CEGMA, which has supplanted our CEGMA results in our revised manuscript. Using the tetrapod gene set, our previous assembly contained 1753/3950 complete BUSCO genes and 2625/3950 complete or fragmented BUSCO genes. Our further scaffolding efforts have yielded an additional 36 complete BUSCO genes, predominantly by completing previously fragmented genes, for a total of 1789 complete and 2647 complete or fragmented BUSCO genes.

Over 22 thousand high confidence protein coding genes were predicted in our draft genome, and we reported putative annotations for nearly 60% of them. Since submission, we have continued this work and have assigned a putative functional annotation to an additional 14.3% through domain homology. These results have been added to the text on page 6. Also, the vast majority of our predicted genes are on scaffolds that include more than 1 kbp of sequence beyond the predicted transcribed region. Prior to this work, to our knowledge there were only 10 Ranid sequences in GenBank that included any amount of promoter sequence.

We propose that this manifold increase in available genomic information for an under-represented genus is of considerable value to the greater genomics community, despite the modest state of the initial draft genome. In the future, through continued efforts and improvements, we expect this draft genome to be succeeded by a reference-grade assembly that is sought by our group and the Reviewer. In the mean time, this initial draft genome and accompanying transcriptome have proven to be an invaluable resource to the community, who until now did not have access to a True Frog genome assembly.

P2. As for the analyses, I think that in general, there needs to be more description of the

rationale and significance of the experiments, especially in regard the differential expression analysis and the non-iridovirus sequence.

R2. We thank the Reviewer for this suggestion. We concur that the evaluation of the non-iridovirus sequence was only tangentially related to the main subject of the manuscript, and we have removed the related text. The predominant rationale for the differential expression experiment was to demonstrate the utility of the gene annotations to genomic and transcriptomic studies with this organism. Additional description of the motivation for the experiments has been added to the text.

P3. Specifically, with regard to the differential expression experiment, the authors speculated that RNA/DNA processing genes predominated because transcripts involved in thyroid hormone-dependent processes require capping and polyA-adenylation. But since all transcripts require that processing, I'm not sure what relevance that explanation has.

R3. Thyroid hormone yields widespread transcriptional change in affected tissues, which are often induced to reprogram cells or mature organs. This is particularly the case for the larval back skin, where exposure to thyroid hormone leads to apoptosis of larval-specific skin cells, and proliferation and differentiation of progenitor cells to form the adult skin. The gene ontologies of the differentially expressed genes are characteristic of an expanding cell population. We regret that the wording of our explanation for this observation implied anything chemically atypical about mRNA molecules from thyroid hormone-regulated genes, and we have rephrased that section of the text for better clarity on page 14.

P4. I was also confused about the amount of description of lncRNAs and their importance to biology since the only relevant data in this paper was a total number of lncRNAs annotated in this assembly.

R4. We thank the Reviewer for their helpful comment, indicating that our description of the significance of the lncRNA results should be clarified. As we have now further emphasized in the text, 1/6th of the differentially expressed transcripts that we detected in the back skin were putative lncRNAs. The sections on pp. 13-14, which discuss known biological functions of lncRNAs and hormonal regulation of lncRNA expression in other organisms, serve to highlight the areas of similarity with the functions of protein-coding genes regulated by thyroid hormone in amphibians. By extension, we posit that these candidate lncRNAs may also act in areas such as transcriptional and translational regulation in the bullfrog. Further, we provide the scientific community our predicted lncRNAs, as noted on p 26.

P5. In addition, I wasn't sure why the authors didn't use a larger dataset than mtDNA to make an amphibian phylogeny since they have so much more sequence they could use. There are salamander chromosomes that have been sequenced, I believe, that could be used for this purpose. And once again, I do not know what conclusions the authors drew from the estimated sequence identity between frogs and toads.

R5. The Reviewer is correct that a small number of amphibians have substantially more annotated genetic sequence available than what we used in our analysis. However, this is only the case for the two *Xenopus* species, *N. parkeri*, and the axolotl, the latter of which consists of only two chromosome gene linkage maps. Restricting our analysis to

these species and the bullfrog would have yielded a much more limited analysis than our choice to use sequence from 15-32 species instead. We feel that this wider variety of species, including toads as outgroups, sufficiently increases the breadth of the analysis to compensate for the relatively shallow depth of sequence from each individual organism.

Smaller points:

P6. Please don't cite timetree.org for divergence times, instead cite the papers that timetree cites

R6. As suggested, we have replaced the in-text citation of timetree.org with the main TimeTree work, since it integrates information from many more focused studies.

REVIEWERS' COMMENTS:

Reviewer #1 (Remarks to the Author):

The authors have satisfactorily addressed all of the concerns that were raised with respect to the first draft.

The only comment I have with respect to this draft is to question whether the modifier "to some extent" is necessary on line 54 of page 3. It seems unnecessary and could be viewed as somewhat offensive by the people that developed those genome resources, at least without a bit of explanation. I should note here that I was not personally involved in either of those studies.

Reviewer #2 (Remarks to the Author):

All of my comments have been well-addressed by the authors.

I have one additional small comment: On line 65, the authors should note that they refer to scaffold NG50.

Reviewers' comments

Reviewer #1:

The authors have satisfactorily addressed all of the concerns that were raised with respect to the first draft.

P1. The only comment I have with respect to this draft is to question whether the modifier “to some extent” is necessary on line 54 of page 3. It seems unnecessary and could be viewed as somewhat offensive by the people that developed those genome resources, at least without a bit of explanation.

R1. We have removed that phrase from line 54, and we thank the reviewer for their helpful comments and suggestions during the review process.

Reviewer #2:

All of my comments have been well-addressed by the authors.

P1. I have one additional small comment: On line 65, the authors should note that they refer to scaffold NG50.

R1. We have added the specification that the NG50 value on line 65 is the scaffold NG50, and express our appreciation of their constructive comments on our manuscript.